# META LEARNING WITH FAST/SLOW LEARNERS

## ABSTRACT

Meta-learning has recently achieved success in many optimization problems. In general, a meta learner $g(.)$ could be learned for a base model $f(.)$ on a variety of tasks, such that it can be more efficient on a new task. In this paper, we make some key modifications to enhance the performance of meta-learning models. (1) we leverage different meta-strategies for different modules to optimize them separately: we use conservative "slow learners" on low-level basic feature representation layers and "fast learners" on high-level task-specific layers; (2) Furthermore, we provide theoretical analysis on why the proposed approach works, based on a case study on a two-layer MLP. We evaluate our model on synthetic MLP regression, as well as low-shot learning tasks on Omniglot and ImageNet benchmarks. We demonstrate that our approach is able to achieve state-of-the-art performance.

## 1 INTRODUCTION

The current success of deep learning hinges on the ability to apply gradient-based optimization in high-capacity models. It has achieved impressive results on many large-scale supervised tasks such as image classification (Krizhevsky et al. (2012); He et al. (2016); Szegedy et al. (2017)) and games (et.al. (2016; 2013)). Notably, although these models achieve superior performance, they require huge amount of data and iterations. In contrast, humans can rapidly learn a novel concept from only a few examples. The ability of fast knowledge acquisition might be related to the meta learning process in the human brain (Harlow (1949)).

To improve learning efficiency, a seminal work named "learning-to-learn", or "meta-learning" (Andrychowicz et al. (2016)) has been proposed to accelerate the learning process. Suppose in a machine learning task, we aim to optimize an objective $f(\theta)$ defined over some domain $\theta \in \Theta$. At step $t$, rather than applying stochastic gradient descent (SGD) as $\theta_{t+1} = \theta_t - \alpha_t \nabla f(\theta_t)$, a meta learner $g$ can be learned to guide the training process of $f$:

$$\theta_{t+1} = \theta_t + g_t(\nabla f(\theta_t), \phi) \tag{1}$$

An interesting phenomenon has been observed in (Andrychowicz et al. (2016)): although successful on MNIST, meta-learning does not perform as well on CIFAR-10. This issue becomes more severe in case of large-scale learning tasks such as ImageNet Training (Russakovsky & et.al. (2015)). To alleviate this issue, a more constrained form of parameter update model is introduced in (Wichrowska et al. (2017)) to improve generalization.

The reason of failure, in our opinion, may lie in that, in a CNN, the low-level convolutional and high-level fully-connected modules play different roles in visual recognition. The former tend to learn generic features across categories, while the latter are generally more task-specific. This phenomenon of feature transferability has been studied in (Yosinski et al. (2014); Azizpour et al. (2016)). If we apply the same learning strategy for all modules, some parameters may not converge well. Accordingly, we explore an improved multiple meta-strategies model. Drawing inspiration from the common trick of "fine-tune" across tasks, we leverage a conservative *"slow-learner"* for low-level modules and a *"fast-learner"* for high-level modules to enable rapid learning.

Notably, another factor enabling fast learning is that human-being could rely on prior knowledge and memories. Low-level perceptional modules could be inheritable characteristics preservable in natural selection over successive generations, undergoing a lifelong learning process (Briscoe & Chittka. (2001)). Accordingly, we adopt a lifelong learning scheme for the slow-learner with parameter shared across new meta-learning tasks. The assumption we make here is that the generic low-level

representation is a good prior "knowledge", which could in turn enables us to learn "smarter". A high level view of our fast/slow meta-learning is shown in Figure 1.

## 1.1 CONTRIBUTIONS

The main contributions of our work are three-folds:

1. We propose a fast/slow meta-learning approach to optimize a convolutional neural network. By applying different learners on different modules, our model could further improve the meta-learning performance method;

2. We provide theoretical study on why the proposed meta-learning on a two-layer MLP;

3. Experiments on the synthetic, Omniglot and ImageNet datasets validate our model.

## 2 RELATED WORK

Early works on *learning-to-learn* or *meta-learning* can be dated back to (Mitchell & Thrun (1993); Hochreiter et al. (2001)). The problem can generally be formulated as a two-level learning task: a meta-level model across tasks and a specific-level model within each task. The former acquires generic knowledge, ideally transferrable to the latter. The specific and meta-level models could be framed as a single learner or separate learners. Along this line, a series of work Vilalta & Drissi. (2002); Baxter. (1998) were proposed. Perhaps, the most general framework is (Schmidhuber. (1993)), which considers a neural network able to modify their own weights.

Recently, a seminal work (Andrychowicz et al. (2016)) leverages one recurrent neural network ("optimizer") to train another base neural network ("optimizee"). The proposed model achieves better learning curves than hand-designed optimizers such as SGD and ADAM (Kingma & Ba (2014)). Our model improves upon their work by explicitly applying fast/slow meta-learners on different modules in the base neural network. (Li & Malik (2017)) relied on policy search to compute meta-parameters. (Wichrowska et al. (2017)) constructed a hierarchical RNN to capture different levels of dependencies.

In particular, meta-learning has shown strong performance in one-shot or low-shot learning tasks, due to its ability to directly generate weights for the base neural network (Woodward & Finn. (2017)). Meta learning with LSTM learners (Ravi & Larochelle. (2017); Munkhdalai & Yu (2017)) and Model-Agnostic learners (Mishra et al. (2017); Finn et al. (2017b;a)) both achieve SOA performance. Additionally, memory-augmented meta-learning (Santoro et al. (2016); Kaiser et al. (2017); Vinyals et al. (2016); Munkhdalai & Yu (2017)) also produce superior result on low-shot benchmarks.

The most related work might be (Munkhdalai & Yu (2017)), where slow/fast weights are also proposed. The main differences are: (1) our model applies lifelong slow learners with parameter sharing; (2) for a specific parameter, we apply either a slow or a fast learner; in contrast, both learners are applied and combined in (Munkhdalai & Yu (2017)). Our model is closely related to MAML (Finn et al. (2017a)) where a good initialized representation is learned and shared across tasks to adapt to new domains.

Meta-learning has also shown promising performance on various tasks such as reinforcement learning (Wang et al. (2017); Duan et al. (2017); Finn et al. (2017b)). It can be even generalized to the scenario where gradients are not available (Chen et al. (2017)).

## 3 META-LEARNING WITH SLOW/FAST LEARNERS

Following (Andrychowicz et al. (2016)), we briefly introduce the notations. Suppose we try to optimize our "optimizee" $f(\theta, \theta')$, where $\theta'$ and $\theta$ stand for different modules in $f$. In our paper we focus on $f$ as a CNN, where $\theta'$ and $\theta$ are bottom convolutional and top fully-connected modules.

We aim to train a pair of optimizers $\theta^*(f, \phi)$ and $\theta'^*(f, \phi')$ parametrized by $\phi$ and $\phi'$. Then, given a distribution of task functions $f$, the expected loss can be formulated as:

$$L(\phi, \phi') = \mathbb{E}_f\{f[\theta^*(f, \phi), \theta'^*(f, \phi')]\} \tag{2}$$

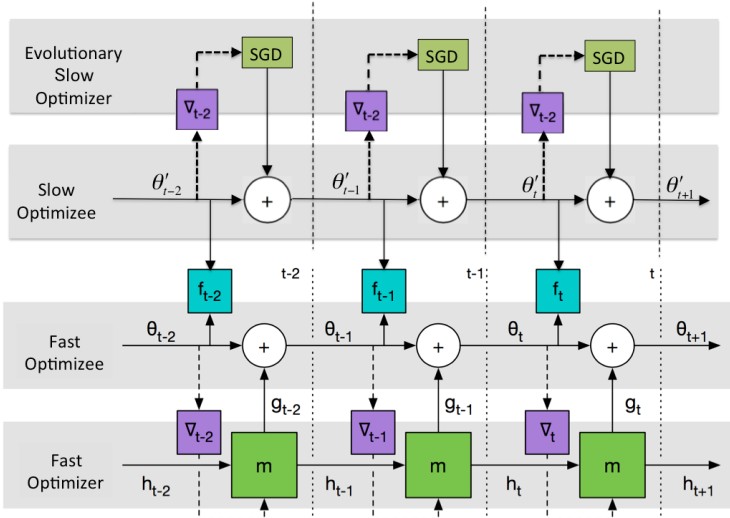

Figure 1: The computational graph of our fast/slow learning model: to optimize the cost function $f(\theta, \theta')$, we apply different strategies on $\theta, \theta'$ separately. We leverage a lifelong-slow learning scheme (SGD) on $\theta'$ to acquire generic knowledge and a fast learning scheme $g(.)$ on task-specific $\theta$. [Photo: inspired by (Andrychowicz et al. (2016))]

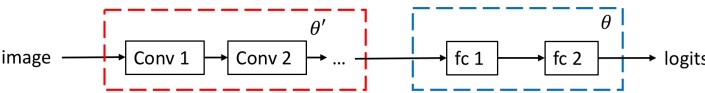

Figure 2: Decoupling the optimization of a CNN into the slow module (generic bottom convolutional layers parametrized by $\theta'$ in the red dashed box) and the fast module (top FC layers parametrized by $\theta$ in the blue dashed box).

We sketch our algorithm in Table.1. A typical setup of our meta-optimizer on a CNN is shown in Figure 2: the modules are decoupled into bottom convolutional $\theta' = \{W_{conv}, ...\}$ and fully-connected ones $\theta = \{W_{fc}, ...\}$. We apply a lifelong-slow strategy on the former, and fast meta-learning on the latter separately. The intuition is that, the more "knowledgeable" $\theta'$ gets, the better the fast meta-learner adapts to a new concept $\theta$. Next, we will carry out a case study on a synthetic shallow network to demonstrate why our approach works.

### 3.1 A CASE STUDY ON TWO-LAYER MLP

As shown in Figure 3, we carry out a case study on a two-layer linear neural network with an $L_2$ regression loss. Bottom and top linear modules are parametrized by $\theta = \{W_f\}$ and $\theta' = \{W_s\}$. We assume that the input $\mathbf{x} \in \mathbb{R}^d$ observes an *i.i.d.* Gaussian distribution $\mathbf{x} \sim N(\mathbf{0}, \Sigma_x)$ with $\Sigma_x$ as a positive-definite matrix; the output $y$ is a scalar. Each time, we create a new linear system task

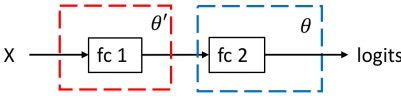

Figure 3: Case study: a two-layer linear MLP with two FC layers.

---

Proposed Fast/Slow Meta Learning Algorithm:

---

Input: a general learning task $f(.; \theta, \theta')$ parametrized by $\theta$ and $\theta'$ with loss function $L$

**Initialize** the slow parameters once $\theta'$;
For epoch $i = 1...n$:
    Sample a new learning task $f_i \in f$; Reinitialize the task-specific parameters $\theta$;
    For $j = 1...m$ iterations:
        1. Sample a small training batch $\{X_j, Y_j\}$;
        2. Calculate batch loss and gradient:
          $\{L_j, \frac{\partial L_j}{\partial \theta}, \frac{\partial L_j}{\partial \theta'}\}$;
        3. Slow and fast module update:
          $\theta_{j+1} = \theta_j + g(\frac{\partial L}{\partial \theta}|_{(\theta_j, \theta'_j)}, \phi)$
          $\theta'_{j+1} = \theta'_j - \delta \frac{\partial L}{\partial \theta'}|_{(\theta_{j+1}, \theta'_j)}$
    Update the parameter $\phi$ in fast learner $g(.)$: $\phi_{i+1} = \phi_i - \delta \frac{\partial \sum_i L_i(\phi)}{\partial \phi}$

---

Table 1: The proposed Fast/Slow Meta-Learning model. The generic slow modules $\theta'$ is only initialized once and shared across tasks. During testing, we fix the parameter $\theta'$ and $\{\phi, \phi'\}$, and apply the learned fast-learner $g(., \phi)$ to update $\theta$.

by sampling a new $\bar{W} \in \mathbb{R}^{1 \times d}$ as our ground truth. We aim to predict $\{W_f, W_s\}$ from noisy labels $y = \bar{W}\mathbf{x} + \epsilon$, where $\epsilon$ is a Gaussian noise with zero mean.

With batch size $k$, we denote the inputs and outputs as $X = [\mathbf{x}_1, \mathbf{x}_2, ..., \mathbf{x}_k]$ and $Y = \bar{W}X + \epsilon = [y_1, ..., y_k]$. Then, denoting the parameters of first and second linear modules as $W_s \in \mathbb{R}^{d \times d}$ and $W_f \in \mathbb{R}^{1 \times d}$, our regression task can be formulated as:

$$L = \frac{1}{k}[\frac{1}{2}\sum_i (W_f W_s \mathbf{x}_i - y_i)^2]$$

and ideally the miracle solution $W_s$ and $W_f$ satisfying:

$$W_f = \bar{W}W_s^{-1}$$

could be learned on a new task $\bar{W}$ as soon as possible. At convergence, we should have:

$$W_f W_s = \bar{W} = (XX^T)^{-1}XY^T = \Sigma_x^{-1}XY^T$$

which is an second-order optimizer. If we are constrained to apply a first-order optimizer (SGD) to optimize $W_s$ and $W_f$, the solution undergoes a zig-zag path and takes many steps to converge, depending on the conditioned number of the covariance matrix $\Sigma_x$.

### 3.1.1 A PERFECT $W_s$ MAKES $W_f$ CONVERGE WITHIN 1-STEP

To simplify notations, we denote $S = W_s^T W_s$ for all the following analysis.

**Lemma 1** *If the following condition satisfies:*

$$S \propto \Sigma_x^{-1}$$

*then suppose we initialize $W_f$ with $\mathbf{0}$, applying one-step SGD on $W_f$ could make the optimization converge s.t. $W_f W_s = \bar{W}$.*

**Proof 1** *The gradient of $L$ with respect to $W_f$ is:*

$$\frac{\partial L}{\partial W_f} = \frac{1}{k}\sum_i (W_f W_s \mathbf{x}_i - y_i)\mathbf{x}_i^T W_s$$

*If we initialize with $W_f = \mathbf{0}$, we have:*

$$\frac{\partial L}{\partial W_f} = -\mathbb{E}\{\mathbf{x}^T W_s^T y\}$$

$$= -\mathbb{E}\{\bar{W}\mathbf{x}\mathbf{x}^T W_s^T\} \quad (\because y = \bar{W}\mathbf{x} + \epsilon)$$

$$= -\mathbb{E}\{\bar{W}\Sigma W_s^T\} = -\mathbb{E}\{\bar{W}\Sigma W_s^T W_s W_s^{-1}\}$$

$$\propto -\mathbb{E}\{\bar{W}W_s^{-1}\} \quad (\because S = W_s^T W_s \propto \Sigma_x^{-1})$$

*The direction of the gradient perfectly aligns with the ground truth solution $W_f = \bar{W}W_s^{-1}$. With an appropriate choice of learning rate for $W_f$, SGD makes $W_f$ converge in one step.*

We can interpret Lemma 1 as: *if we have a miracle representation $h = W_s^* x$ as a whitening operator, the top module $W_f$ can be learned by SGD within one step, regardless of the task target $\bar{W}$.*

### 3.1.2  FAST/SLOW META-LEARNING MAKES $W_s$ CONVERGE TO MIRACLE

Since a miracle $W_s$ (i.e., $W_s$ is the whitening matrix of input $x$) makes $W_f$ adapt to new tasks, it is of interest if we could learn the miracle $W_s^*$? Next, we demonstrate that the proposed fast/slow meta-learning model could converge to the good solution.

Following previous analysis, the gradient of our top module $W_f$ with respect to loss $L$ given a batch $\{X_1, Y_1\}$ of batch size $k$ is:

$$\frac{\partial L}{\partial W_f} = \frac{1}{k}(W_f W_s X_1 - Y_1)X_1^T W_s^T$$

Suppose we always initialize $W_f = \mathbf{0}$. Then suppose the fast-learner is a SGD with learning rate $\eta$, we have the updated $W_f$ as:

$$W_f = \frac{\eta}{k}Y_1 X_1^T W_s^T \tag{3}$$

To improve generality, we ideally expect the new $W_f$ perform well on a new sampled batch $\{X_2, Y_2\}$:

$$L = \frac{1}{2}(W_f W_s X_2 - Y_2)^2$$
$$= \frac{1}{2}(\frac{\eta}{k}Y_1 X_1^T W_s^T W_s X_2 - Y_2)^2 \qquad (\because \mathbf{Eqn}(3))$$
$$= \frac{1}{2}(\frac{\eta}{k}Y_1 X_1^T S X_2 - Y_2)^2 \qquad (\because W_s^T W_s = S)$$

**Lemma 2** *We denote $A = \mathbb{E}(\bar{W}\bar{W}^T)$ as the expectation of weight covariance matrix across tasks. If we follow the fast/lifelong-slow meta-learning strategy, at convergence $S = W_s^T W_s$ satisfies:*

$$S^{-1} = \eta(k+1)(\Sigma_x + \frac{\mathbf{tr}(\Sigma_x + A) + \epsilon^2}{k+1}A^{-1}) \tag{4}$$

*where $k$ is the batch size of $\{X_2, Y_2\}$.*

**Proof 2**

$$\mathbb{E}(\frac{\partial L}{\partial S}) = \eta\mathbb{E}\{X_1 Y_1^T(\frac{\eta}{k}Y_1 X_1^T S X_2 - \bar{W}X_2 - \epsilon)X_2^T\}$$
$$= \eta\mathbb{E}\{X_1 Y_1^T(\frac{\eta}{k}Y_1 X_1^T S - \bar{W})X_2 X_2^T\}$$
$$= \eta\mathbb{E}\{X_1(X_1^T \bar{W}^T + \epsilon)(\frac{\eta}{k}(\bar{W}X_1 + \epsilon)X_1^T S - \bar{W})\}\Sigma_x$$
$$= \frac{\eta^2}{k}\mathbb{E}\{X_1(X_1^T \bar{W}^T \bar{W}X + \epsilon^2)X_1^T)S\Sigma_x\} - \frac{\eta}{k}\mathbb{E}\{X_1 X_1^T \bar{W}^T \bar{W}\}\Sigma_x$$
$$= \eta((k+1)\Sigma_x A\Sigma_x) + (\mathbf{tr}(\Sigma_x A) + \epsilon^2)\Sigma_x)S\Sigma_x - \eta\Sigma_x A\Sigma_x \qquad (Lemma-3)$$
$$= \eta\Sigma_x A(\eta((k+1)\Sigma_x + (\mathbf{tr}(\Sigma_x A) + \epsilon^2)A^{-1})S - I)\Sigma_x$$

*At convergence, an equilibrium should be reached where $\mathbb{E}\{\frac{\partial L}{\partial S}\} = 0$, then:*

$$S^{-1} = \eta(k+1)(\Sigma_x + \frac{\mathbf{tr}(\Sigma_x + A) + \epsilon^2}{k+1}A^{-1})$$

In the proof, we need the following lemma:

**Lemma 3** *If* $\mathbf{x} \sim N(\mathbf{0})$*, then for a matrix A, we have:*

$$\mathbb{E}(\mathbf{x}\mathbf{x}^T A \mathbf{x}\mathbf{x}^T) = \Sigma_x A \Sigma_x + \Sigma_x A^T \Sigma_x + \mathbf{tr}(\Sigma_x A)\Sigma_x \tag{5}$$

*and if* $X = [\mathbf{x}_1, \mathbf{x}_2, ...]$ *with each column sampled from an i.i.d. Gaussian* $\mathbf{x}_i \sim N(\mathbf{0}, \Sigma_x)$*, then:*

$$\mathbb{E}(XX^T A XX^T) = k(k+1)\Sigma_x A \Sigma_x + \Sigma_x A^T \Sigma_x + k\mathbf{tr}(\Sigma_x A)\Sigma_x \tag{6}$$

**Proof 3** *The moment-generating function of* $\mathbf{x}$ *is* $\phi(\mathbf{t}) = \exp(\frac{\mathbf{t}^T \Sigma_x \mathbf{t}}{2})$*, then for index* $\{i, j, k, l\}$ *we have:*

$$\mathbb{E}(x_i x_j x_k x_l) = \frac{\partial^4 \phi(\mathbf{t})}{\partial t_i \partial t_j \partial t_k \partial t_l} = \sigma_i \sigma_j \sigma_k \sigma_l \tag{7}$$

*Hence,*

$$\mathbb{E}(\mathbf{x}\mathbf{x}^T A \mathbf{x}\mathbf{x}^T) = \sum_{k,l} \mathbb{E}(x_i x_k A_{kl} x_l x_j)$$

$$= \sum_{k,l} A_{k,l}(\sigma_{i,j}\sigma_{k,l} + \sigma_{i,k}\sigma_{j,l} + \sigma_{i,l}\sigma_{j,k})$$

$$= [\mathbf{tr}(\Sigma_x A)\Sigma_x]_{i,j} + [\Sigma_x A \Sigma_x]_{i,j} + [\Sigma_x A^T \Sigma_x]_{i,j}$$

*and*

$$\mathbb{E}(XX^T A XX^T) = (\sum_{i=1}^{k} \mathbf{x}_i \mathbf{x}_i^T) A (\sum_{j=1}^{k} \mathbf{x}_j \mathbf{x}_j^T) = \sum_{i=1}^{k} \sum_{j=1}^{k} \mathbf{x}_i \mathbf{x}_i^T A \mathbf{x}_j \mathbf{x}_j^T$$

$$= k(k-1)\Sigma_x A \Sigma_x + 2k\Sigma_x A \Sigma_x + k\mathbf{tr}(\Sigma_x A)\Sigma_x$$

$$= k(k+1)\Sigma_x A \Sigma_x + k\mathbf{tr}(\Sigma_x A)\Sigma_x$$

## 3.2 THE INSIGHT OF THE CASE-STUDY

The above analysis provides why the fast/slow meta-learning works:

1. A *miracle* knowledge of input signal $x$ (i.e., $W_s$ whitens $x$), makes top module $W_f$ learnable in one SGD step for a new task; (Lemma 1)

2. Alternating lifelong-slow $W_s$ and fast $W_f$ learning, we are able to learn $W_s$ close to the miracle knowledge of $x$ with a regularization term. (Lemma 2)

## 4 EXPERIMENT

We evaluate our algorithm on three datasets: synthetic two-layer neural network, the Omniglot low-shot benchmark and the large-scale ImageNet dataset.

### 4.1 SYNTHETIC DATASET

Our first experiment is to verify our theoretical analysis on the two-layer MLP as mentioned in the last section. We set the dimension as $d = 10$, such that input data $\mathbf{x} \in \mathbb{R}^{10}$, each sampled new task has a ground truth $\bar{W} \in \mathbb{R}^{1\times10}$, $W_s \in \mathbb{R}^{10\times10}$ and $W_f \in \mathbb{R}^{1\times10}$. The input $x$ is sampled from an *i.i.d.* Gaussian $\mathbf{x} \sim N(\mathbf{0}, \Sigma_x)$ with $\Sigma_x$ fixed across all tasks; each new task is sampled from another Gaussian $\bar{W} \sim N(0, I)$. For each task, we set the batch size as $k = 100$.

For the slow learner $g'(.)$, we use a conservative SGD with the learning rate $\delta = .001$; for the fast learner $g(.)$, we use a large-step SGD with a learning rate $\eta = 0.2$. In Figure 4, we plot the learning-curve of the regression problem. As shown in Figure 4 (a), we show the performance gain defined as the relative error of 1-step SGD update: $\frac{L_1}{L_0} = \frac{||Y - W_s W_{f,1} X||^2}{||Y - W_s W_{f,0} X||^2}$. We can see that as the slow learner gradually captures the distribution of input signal $x$, the fast learner could be very efficient ($L_1/L_0$ decreases from 0.8 to 0). In Figure 4 (b), we evaluate how well $W_s$ captures the knowledge of $x$ throughout our fast/slow meta learning process. The metric we use is the Frobenius-norm between $W_s^T W_s$ and the true whitening matrix: $||(W_s^T W_s) - \frac{1}{\eta}\Sigma_x^{-1}||_{Frob}^2$ We can see clearly that

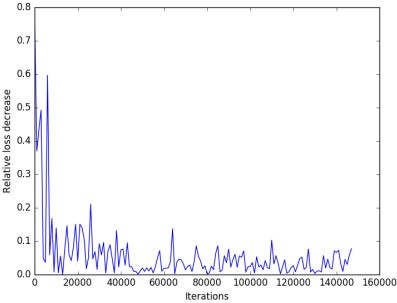 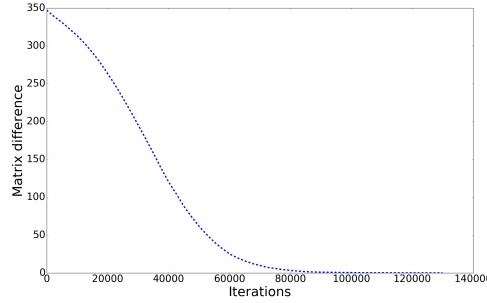

(a) Relative Training Loss of 1-Step Fast Update

(b) Estimation Loss bottom module between the learned and optimal $W_s$

Figure 4: Experiments on the two-layer linear MLP. (a) x-axis: fast/slow training epochs; vertical-axis: Relative loss $L(W_{f,1})/L(W_{f,0})$ after one-step fast module update with a large learning rate $\eta = 0.2$; (b) x-axis: fast/slow training epochs; vertical-axis: Frobenius-norm of the estimation difference between slow module $\theta' = \{W_s\}$ and the true whitening precision matrix $\Sigma_x^{-1}$.

$W_s^T W_s$ approach the true precision matrix $\Sigma^{-1}/\eta$, i.e., the slow learner evolves to capture the input distribution "magically" after 100,000 iterations.

**Summary**: The experiments aligns perfectly with our theoretical analysis in above sections: the combination of a slow learner and a fast learner achieves state-of-the-art performance.

## 4.2    VISUAL RECOGNITION

We briefly introduce the setup of our fast/slow meta-learning on visual classification tasks. We divide the datasets into 3-split: on $C_{base}$ classes, we first pre-train a base CNN $f(\theta, \theta')$ ; then on $C_{meta}$, we train our fast meta-learner $g(\phi)$ to generate $\theta$ and fine-tune our lifelong slow parameters $\theta'$ with a slow-learner (SGD); on $C_{test}$ we test our model. For our fast learner $g(., \phi)$, we use a coordinate-wise LSTM with 10-steps.

To evaluate our approach in comparison to prior meta-learning and low-shot visual algorithms, we apply our algorithm on the Omniglot benchmark (Lake et al. (2011)) and MiniImageNet, which are most common recently-used low-shot benchmarks. Omniglot contains $C_{base} = 964$ characters in the training set, and we divide the 1,200 categories in the testing set into $C_{meta} = 600$ and $C_{test} = 600$. The MiniImagenet task proposed by (Ravi & Larochelle. (2017)) contains 100 categories with $C_{base} = 64$, $C_{meta} = 16$ and $C_{test} = 20$ respectively. We follow the experimental protocol of (Vinyals et al. (2016)) such as multiple 90 degree rotation for Omniglot.

For a fair comparison, we follow the same model architecture as in Matching Network (Vinyals et al. (2016)) and MAML (Finn et al. (2017a)). For Omniglot, we use a simple yet powerful CNN, containing 4 modules with $3 \times 3$ convolution with $64$ filters followed by batch-normalization (Ioffe & Szegedy (2015)), ReLU and $2 \times 2$ max-pooling. For MiniImagenet, we use 32 filters per layer to avoid overfitting as done in (Ravi & Larochelle. (2017); Finn et al. (2017a)). A top fully-connected layer $\theta = \{W_{fc}\}$ followed by a softmax non-linearity is applied to define the Baseline Classifier.

The performance comparison of different approaches is shown in Table 2. Our approach is highly competitive with the State-Of-the-Art (SOA) on both Omniglot and MiniImagenet.

**FullImageNet**:
Finally, it is of interest to test how our algorithm combined with SOA powerful CNN on the most challenging real-life scenarios– Full ImageNet. We use the Inception-ResNet-V2 model (Szegedy et al. (2017)) as our base model. The task contains 21,841 categories (non-empty synsets) and the total number of images up to 14,197,122. Generally, the classification performance is tested on the 1,000 base classes, with 1,280,000 images. We adopt as our base CNN model and test our algorithm in the one-shot scenario on the remaining 20,841 unseen categories. We use half of the data for meta-learning training and the other half for testing.

| Model | Omniglot (20-way, 1-shot) | MiniImagenet (5-way, 1-shot) |
|---|---|---|
| Pixel-KNN | 26.7% | - |
| MANN (Santoro et al. (2016)) | 36.4% | - |
| CNN (Our implementation) | 85.0% | 28.8% |
| Siamese (Koch et al. (2015)) | 88.1% | 40.1% |
| Feature Shrinking (Hariharan & Girshick (2017)) | 89.5% | 44.0% |
| Matching Network (Vinyals et al. (2016)) | 93.8% | 46.6% |
| Learning to remember (Kaiser et al. (2017)) | 95.0% | - |
| MAML (Finn et al. (2017a)) | 95.8% | 48.7% |
| Prototypical Net (Snell et al. (2017) | 96.0% | - |
| Meta Networks (Munkhdalai & Yu (2017)) | 97.0% | - |
| **Temporal Conv (Mishra et al. (2017))** | **97.6%** | - |
| **Fast/Slow (Ours)** | **96.8%** | **48.8%** |

Table 2: Performance comparison of our algorithms and the state-of-the-art on Omniglot and Mini-ImageNet.

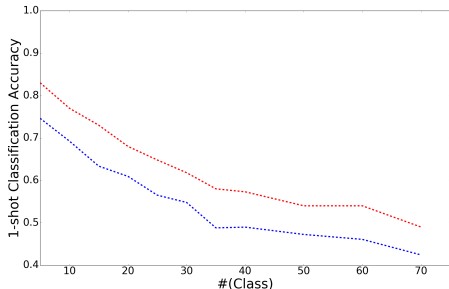

Figure 5: FullImageNet: 1-shot Classification Accuracy w.r.t. class number. Red line: our meta-learning; blue line: CNN metric-learning.

As shown in Figure 5, we plot the 1-shot classification accuracy on the novel classes with respect to class number (for example, $|class| = 10$ means we carries out a 1-shot 10-way task). We can see our algorithm (red) steadily outperforms the strong baseline (CNN with metric learning).

## 5 CONCLUSION

In this paper, we propose a fast/slow meta-learning approach to optimize a base convolutional neural network. By applying different strategies on bottom-generic/top-task-specific modules, our approach further improve the meta-learning performance. Theoretical analysis on a two-layer MLP regression problem, and extensive experiments on the synthetic, Omniglot one-shot, as well as ImageNet benchmarks validate the effectiveness of our approach.

## 6 LIMITATIONS AND FUTURE WORK

A strong assumption made in our theoretical analysis is that the input data observes the same distribution across tasks $x \sim N(\mathbf{0}, \Sigma)$. However, this does not necessarily hold true. In case the data from some new categories observe a different distribution, we might further extend our model to a hierarchy:

$$x_i \sim P_z(\theta') \qquad z \in \{1, ..., m\}$$

where we assume the input signal $x_i$ observes a mixture model and the latent variable $z$ could be regarded as part of the slow module $z \in \theta'$. In the future, we will explore how to extend our model to the scenario of a mixture model.

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
