# OpenReview forum: "Meta Learning with Fast/Slow Learners"
_ICLR.cc/2019/Conference_

### Official Review · AnonReviewer1 · 2018-11-02
**Training slow and fast learners using different strategies is an interesting idea.**

**Rating:** 5
**Confidence:** 4

**Review:**

[Summary:]
This paper presents a meta-learning architecture where the slow learner is trained by SGD and the fast learner is trained according to what the meta-learner guides. CNN is split into two parts: (1) bottom conv layers devoted to learn meaningful representation, which is referred to as slow learner; (2) top-fully connected layers involving task-specific fast learners. As in [Andrychowicz et al., 2016], the meta-learner guides the training of task-specific learners. In addition, slow learners are trained by SGD. The motivation is that low-level features should be meaningful everywhere while high-level features should vary wildly. They introduce “miracle representations” and prove that fast/slow learning on a two-layer linear network should converge to somewhere near this miracle representation. They evaluate on few-shot classification benchmarks to evaluate how well this fast/slow meta-learning approach works.

[Strengths:]
The paper has a clear motivation. It is easy to read. Training slow/fast learners using different strategies is an interesting idea.

[Weaknesses:]
- The technique used in this work is a mix of SGD and  [Andrychowicz et al., 2016].
- The analysis is limited to a simple two-layer linear network. It is not clear whether this analysis is carried over to the proposed deep nets.
- Quantitative results did not compare to recent results such as Reptile[1] or MT-Nets[2].

[Specific comments:]
- The current work is an improvement over [Andrychowicz et al., 2016], claiming that training conv layers and fully-connected layers with different strategies improves the generalization. I am wondering why the comparison to [Andrychowicz et al., 2016] is missing. You can use (fully) pre-trained CNN (which already learns meaningful representation using a huge amount of data) in the framework of [Andrychowicz et al., 2016].
-As one of the points of the paper is that this meta-learning strategy enables life-long learning, it would have been nice to see an experiment using this, where the distribution of tasks changes as time goes on.
-The paper says SOA(State Of the Art); I think the term SOTA(State Of The Art) is more commonly used.
-The use of the term “miracle” keeps changing(miracle solution, miracle representation, miracle W, miracle knowledge); the paper would be clearer if only one “miracle X” was defined and used as these are all essentially saying the same thing.

References
[1]https://arxiv.org/abs/1803.02999
[2]https://arxiv.org/abs/1801.05558

---

### Official Review · AnonReviewer3 · 2018-11-02
**An interesting treatment to meta-learning with fast/slow learning components.**

**Rating:** 6
**Confidence:** 3

**Review:**

[Summary]
The paper presents a novel learning framework for meta-learning that is motivated by neural learning process of human over long periods. Specifically, the process of meta-learning is divided into a slow and a fast learning modules, where the slowly-learnt component accounts for low-level representation that is progressively optimized over all data seen so far to achieve generalization power, and the fastly-learnt component is supposed to pick up the target in a new task for quick adaptation. It is proposed that meta-learning should focus on capturing the meta-information for the fast learning module, and leave the slow module being updated steadily without task-specific adaptation. Theoretical analysis is presented on a linear MLP examples to shed some light on the properties of the proposed algorithm. Results on both synthetic dataset and benchmarks justify the theoretical observation and advantages.

Pros
Novel treatment and formulation of meta-learning from the perspective of fast and slow  learning process
Cons
Some interesting cases not tested
Presentation could be improved

[Originality]
The paper approaches the recently popular meta-learning from a novel perspective by decomposing the learning process into slow and fast ones.

[Quality]
Overall,  the paper is well motivated and implemented with both theoretical study and empirical justification. There are a few questions / areas for further improvements, though:
- It seems that to initialize the slow module, another set of data is needed to pretrain it before the actual meta-learning takes place to learn to optimize the fast learner (as opposed to other meta-learning methods where all parameters in a base model were meta-learnt over the meta training set). How does this affect the performance? E.g., what if the slow module is only updated over the meta-training set (still without reinitialization across different batches) without pre-training?
- In the current formulation, the base model is decomposed into two distinct (slow and fast) modules. What is the rule to decide which layers should belong to slow or fast modules? How does different choice affect the performance? Can we decompose the base model into finer granularities for different learning behaviors? E.g., a third module module in-between the fast and slow ones that follows medium learning pace.
- The theoretical study can be better organized. The proofs can be left in appendix to make room for more discussion on conclusions, non-linear and / or non-Gaussian cases.
- The write-up can be improved too at some places: proper reference at line 4 of section 1 is missing; \phi in (1) is not well defined, as well as “SOA” in section 2;

[Clarity]
The paper is generally clearly written, with a few places to improve (see comments above).

[Significance]
The paper brings in an interesting perspective to meta-learning. It can also inspire more follow-up work to better understand the problem.

---

### Official Review · AnonReviewer2 · 2018-11-02
**The paper addresses fast/slow learning modules in deep networks. Good paper. Needs more clarity/work.**

**Rating:** 5
**Confidence:** 3

**Review:**

The overall contribution makes sense. Consider solving a linear system i.e., learning an unknown matrix. Splitting it into two components (like in NMF or MMF) and learning each separately gives more control on the conditioning of the matrices. This is the basis of residual networks (at least the theory for linear resnets). Within this, the technical/theoretical results presented in the paper are sensible. Couple of issues:
1) Where are we breaking the slow/fast learners in terms of the depth of the network? I.e., How many of the layers are slow? Does this break point influence the overall convergence?
2) It is unclear what the aim of simulations is? The reported figures are not conveying useful information. It makes sense to do a repeatability experiment here with multiple sets of simulated datasets.
3) Put confidence intervals on the results (table/figure).
4) What is the nature and choice of g()? The evaluations uses LSTM but will the structure of g() influence the rate of learning?
5) The authors should choose a better reference than miracle for the

---

### Meta-Review · Area_Chair1 · 2018-12-14
**Useful idea, requires more thorough experiments**

**Confidence:** 5
**Recommendation:** Reject

**Metareview:**

The paper introduces an interesting idea of using different rates of learning for low level vs high level computation for meta learning. However, the experiments lack the thoroughness needed to justify the basic intuition of the approach and design choices like which layers to learn fast or slow need to be further ablated.